# OpenReview forum: "Adversarial Training of Reward Models"
_colmweb.org/COLM/2025/Conference — COLM 2025_

### Official Review · Reviewer_afoe · 2025-04-21

**Rating:** 6
**Confidence:** 3
**Ethics Flag:** 1

**Summary:**

This paper introduces a novel adversarial training of reward models. The methodology involves the following key steps: Adversarial Policy Training; Data Filtering, Adversarial Training, and Iterative Refinement. Using both synthetic and real data sets, authors demonstrate high attack success rate and a slightly higher score compared to conventional training.

**Questions To Authors:**

Q1. In Eq. 5, is y_2 also involved? If not, why? Is it equivalent to replace y_1 with y_2? Why can Z-score be used with only one preference pair?  Why is Z-score cutoff set to -1.96 not even more negative values, say -2.95?

Q2. Figure 2 gives three adversarial examples from three models and describes characteristics. How are these examples be selected? Any explanation why different models learn so different adversarial examples?

Q3. What are the real-world use cases of this type of attack? Any defense strategies?

Q4. How is Win Rate defined in Figure 5?

**Reasons To Accept:**

1. It appears to be one of the first adv training of reward model in literature.

2. The adv examples are shown to be highly effective in terms of successful rate.

**Reasons To Reject:**

1. The definition of attack success is not clearly described/justified (see question 1 below).

2. The definition of downstream policy performance is not entirely clear (see question 4 below)

3. The real-world use cases and threat model is not clear (see question 3 below).

4. The literature review is somewhat limited in scope (i.e., section on "Adversarial Attacks on LLMs"). For example, prompt injection attack finds applications to jailbreaking can be relevant.

---

> ### Author Response · Authors · 2025-05-30
> **Response to Reviewer afoe**
>
> Dear Reviewer afoe,
>
> Thank you for the detailed and thoughtful review! We are glad you appreciated our work's novelty and effectiveness. We believe that your critiques can be cleared up by improving the writing. We leave our rebuttal to your review below.
>
>
> **Reason to Reject 1: See Q1**
>
> **Reason to Reject 2: See Q4**
>
> **Reason to Reject 3: See Q3**
>
> **Reason to Reject 4: Literature review.**
> - Thank you for the suggestion! We will add discussion on prompt injection attacks to the related work section. However, we remark that prompt-injection attacks are not directly applicable to RMs. Prompt-injection attacks try to add some text to the prompt so that the policy model acts against its guidelines. For RM attacks, we need to have two prompt-response pairs such that one prompt-response pair is better than the other, but the RM thinks it is worse. It is not clear how prompt-injection attacks could be applied to our setting. The key way to think about this is that we are trying to attack a scoring model, not a generative policy model.
> - We do include some simple attack strategies that are designed to attack classifiers, Textfooler and StyleAdv. However, neither of these work as well as AdvRM.
>
>
> **Question 1: On equation 5.**
> - In equation 4, we sample $y_2$ from the SFT model. However, this sampling can be quite noisy and make our evaluation noisy. To reduce the noise, we instead compare the score of $y_1$ to the mean score of samples $y_2$ from the SFT model for the given prompt. In essence, we are saying the sample $y_1$ must have higher $R_1$ compared to the average SFT sample, while having much lower $R_2$ than the average SFT sample. This is a more strict requirement, and also reduces noise by using the mean operation. Note we can recover an attack pair ($y_1$, $y_2$) by picking $y$ sampled from the SFT model with reward similar to the mean SFT sample reward.
> - We can use a Z-score with just one sample $y_1$, because we are comparing $y_1$ with a set of responses y sampled from the SFT distribution (usually 4 or 8). We will clarify this in the next version of the paper.
> - The z-score cutoff is set to -1.96 as that is a very common threshold in scientific literature. Lower thresholds could be used as well.
>
> **Question 2: Adversarial examples characteristics.**
> - These samples are randomly sampled from attacks that meet our filtering criteria (see line 172). We do not have a concrete explanation for why different attacks are learned for different models, but we believe it is due to different training data, training strategies, and pre-trained base models being used for the different reward models.
>
> **Question 3: Practical use cases and defense strategies.**
> - The most straightforward use of these attacks is to further train reward models, which we show makes them perform better in downstream RLHF tasks (Figures 4 and 5) and makes them become more robust (Figure 6a). We discuss this use case in depth in Section 5.
> - Another use case of AdvRM is for red-teaming reward models that are used in moderation systems [1]. In particular, AdvRM could be used to ensure no harmful outputs are seen as harmless by the system. We will discuss this use case in the paper.
> 1. Inan et al., "LLaMA Guard: LLM-Based Input-Output Safeguard for Human-AI Conversations", arXiv (2023)
> - As far as defense strategies, we are not aware of any existing ones. We propose AdvRM training to defend against AdvRM attacks (which is shown to improve robustness in Figure 6a). We hope our work will encourage others to work on RM attack and defense strategies.
>
> **Question 4: Win rate definition**
> - Win rate is defined as #win / (#win + #lose), where the different LLMs or humans are used as judges. We compare responses generated by the trained models to those from the SFT model, and ask the LLM judge to compare them in both orders (SFT response first and SFT response second). This evaluation setup is consistent with the majority of LLM alignment works [1,2,3]. We will clarify this in our experiments section. Note we include details on our evaluation setup in Appendix G.
> 1. Rafailov et al., "Direct Preference Optimization: Your Language Model Is Secretly a Reward Model", NeurIPS 2023
> 2. Chen et al., "Alpagasus: Training a Better Alpaca with Fewer Data", arXiv (2023)
> 3. Zhao et al., "SLIC-HF: Sequence Likelihood Calibration with Human Feedback", arXiv (2023)
>
>
> Please let us know if you have any further questions! We believe by adding your suggestions, we will improve our paper. We look forward to further discussion.

---

> > ### Comment · Reviewer_afoe · 2025-06-05
> >
> > The rebuttal was well-written and clarified some points, but it did not significantly alter my overall assessment. I am keeping my score unchanged.

---

> > > ### Author Response · Authors · 2025-06-06
> > > **Thank you for the discussion!**
> > >
> > > Thank you again for the detailed review! Please let us know if you have any further questions.

---

### Official Review · Reviewer_ALVK · 2025-05-08

**Rating:** 6
**Confidence:** 3
**Ethics Flag:** 1

**Summary:**

This paper introduces an adversarial training framework called Adv-RM, which aims to enhance the robustness of reward models to prevent exploitation in reinforcement learning. By generating adversarial examples and incorporating them into the training process of reward models, the framework significantly improves the robustness of these models. The effectiveness of Adv-RM is demonstrated in both synthetic and real-data settings.

**Questions To Authors:**

Please refer to "Reasons To Reject"

**Reasons To Accept:**

1. The authors propose that uncertainty estimation has limitations in measuring the quality of in-distribution responses, especially under low uncertainty conditions, where the quality signals it provides are very limited. This finding offers a new perspective for assessing the robustness of reward models, reveals the shortcomings of existing methods, and points the way for future research.
2. The authors' proposed Adv-RM framework innovatively generates adversarial examples to address the vulnerability of reward models to exploitation by out-of-distribution (OOD) samples. It significantly reduces the risk of reward hacking and enhances the security of language model applications. The experimental results robustly validate its effectiveness and performance improvements.

**Reasons To Reject:**

1. Although the authors have elaborated on the effectiveness of adversarial examples in theory and conducted extensive experimental validation in synthetic and real-data settings, could they consider further expanding the scope of experiments in certain specific experimental setups (for example, different types of out-of-distribution samples or more complex language models) to more comprehensively assess the robustness and applicability of the Adv-RM framework?
2. In Figure 5(a) and Figure 6(c), the icon partially obscures the content of the image, which may affect the readability and clarity of the chart. It is recommended to make appropriate modifications to enhance the readability of the chart.

---

> ### Author Response · Authors · 2025-05-30
> **Rebuttal To Reviewer ALVK**
>
> Dear Reviewer ALVK,
>
> Thank you for the helpful review. We sincerely believe both of your critiques are addressable. We address your critiques below.
>
> **Reason to reject 1: Scope of experiments.**
> - We agree it is important to evaluate the effectiveness of AdvRM on different types of OOD data. However, as this is one of the first works to attack RMs, there are limited types of OOD samples we can try. In the paper we try Textfooler (which does not work well on RMs), StyleAdv, and transfer attacks from hacking other RMs. Recently, Rewordbench[1] showed how we can perturb some existing benchmarks to attack RMs, but the authors do not publicly release their data or code. However, if any other new attack method comes out, we will try it with our method. In addition, we show AdvRM improves downstream RLHF performance, which shows the widespread applicability of AdvRM. We will add the need for more adversarial attacks on RMs as in the discussion section.
>
> 1. Wu et al., "reWordBench: Benchmarking and Improving the Robustness of Reward Models with Transformed Inputs", arXiv 2025
>
> - As far as using more complex language models, we remark that we experiment with very powerful and complex RMs, including those with 340B parameters.
>
>
>
> **Reason to reject 2: Chart readability.**
> - Thank you for pointing this out. We will fix this plot in the next version of the paper.
>
> Please let us know if you have any other questions! We look forward to further discussion.

---

> > ### Author Response · Authors · 2025-06-06
> > **Thank you for the discussion!**
> >
> > Thank you again for the insightful review! Please let us know if you have any further questions.

---

> ### Comment · Reviewer_ALVK · 2025-06-05
>
> I have read others' comments and the authors' response, I insist on not chaning my score.

---

### Official Review · Reviewer_qJrb · 2025-05-12

**Rating:** 8
**Confidence:** 4
**Ethics Flag:** 1

**Summary:**

This paper introduces Adv-RM, a novel adversarial training framework designed to improve the robustness of reward models (RMs) used in aligning LLMs via RLHF. The core problem addressed is that RMs often assign high rewards to low-quality OOD samples, leading to reward hacking where the LLM policy exploits these RM weaknesses rather than genuinely aligning with the true human preferences.
Adv-RM tackles this by first training an adversarial policy using reinforcement learning. This policy learns to generate responses that:
- Receive a high reward score from the target RM ($R_{θ_1}$).
- Are considered OOD, as measured by high disagreement between the target RM ($R_{θ_1}$) and another competent RM ($R_{θ_2}$). Specifically, the adversarial policy maximizes $R_{θ1}(x,y) - λR_{θ_2}(x,y)$ while ensuring $R_{θ1}(x,y)$ is above a threshold that varies given the setting (synthetic vs real RLHF).
- Are of low actual quality despite the high ${R_θ1}$ score.

The generated adversarial examples ($y_{adv}$) are then filtered and incorporated into the RM training data as rejected samples. They are paired with high-quality responses from a supervised fine-tuned model (y_SFT) to form new preference pairs ($y_{SFT} > y_{adv}$). Retraining the RM on this augmented dataset results in an "Adv-RM" that is more robust.
The authors demonstrate empirically that their method can effectively find vulnerabilities in state-of-the-art RMs ( Skywork-Gemma-27B, Llama-Nemotron-70B, Nemotron-340B). RMs trained with Adv-RM show increased stability during downstream RLHF, mitigating reward hacking and enabling longer, more effective policy training in both synthetic and real-data settings. While the computational cost is higher (approx. 3x standard RLHF), the gains in robustness and training stability are presented as significant.

**Questions To Authors:**

- When attacking Llama-3.1-Nemotron-70B-Reward, you used an 'in-house RM' as $R_{θ2}$. Could you elaborate on its characteristics (architecture, size, training data)? More broadly, how critical is the choice of $R_{θ2}$ to the success and nature of the discovered vulnerabilities, and how sensitive are the results if $R_{θ2}$ significantly differs from $R_{θ1}$ in capacity or training?
- The paper shows Adv-RMs become robust to your specific attack method. Have you evaluated or do you have intuitions on how well this robustness generalizes to other, unseen types of OOD inputs or different reward hacking strategies not explicitly encountered during Adv-RM training?
- You found attacks were no longer successful after 2 rounds of adversarial training. Was this primarily because $\pi_{adv}$ couldn't find new vulnerabilities against the hardened Adv-RM, or did the Adv-RM become so robust that even OOD samples found by $\pi_{adv}$  no longer received high scores? What does this imply about the limits of this iterative hardening process?
- The 3x computational cost is significant. Based on your experiments, do you have insights on whether a 'lighter' or more cost-effective version of Adv-RM could still offer a substantial portion of the robustness benefits, and what might be the trade-offs?
- The adversarial objective $R_{θ1} - λR_{θ2}$ aims for disagreement. Beyond high $R_{θ1}$ scores and low $R_{θ2}$ scores, what specific types of 'low quality' did the generated $y_{adv}$ predominantly exhibit (e.g., factual inaccuracies, incoherence, harmfulness, stylistic issues)? Did $R_{θ2}$ consistently penalize particular failure modes that $R_{θ1}$ missed?
- Beyond improving robustness, do you see a role for the adversarial policy and the generated examples $y{adv}$ as a systematic tool for diagnosing failure modes or 'blind spots' of existing RMs?

**Reasons To Accept:**

I believe this paper warrants acceptance due to its strong contributions across several key dimensions:
- Addresses an important and timely problem: reward hacking is a fundamental obstacle to reliable LLM alignment and improving RM robustness is necessary for progress, making this work highly relevant and impactful.
- Novel and Principled Method: the approach of training an RL policy to generate targeted adversarial examples for RMs, by maximizing disagreement while maintaining high target RM scores, is a clever and well-reasoned advancement over simpler perturbation or random data augmentation techniques. The formulation of the adversarial reward and the subsequent data filtering are sound.
- Strong Empirical Validation. The paper provides compelling evidence of Adv-RM's effectiveness through:
1) Successful attack generation, demonstrates high success rates in finding vulnerabilities in SOTA RMs, supported by clear qualitative examples (e.g., Figure 2) and quantitative metrics (Tables 1 & 2) across multiple judges.
2) Improved downstream performance, showing significant improvements in RLHF training stability (figures 4 & 5) and allowing policies to train for longer without succumbing to reward hacking, leading to more performant aligned models.
3) Thorough evaluation, including comprehensive comparisons against relevant baselines, insightful ablation studies, and experiments in both synthetic and real-data settings.
- Clear insights and analysis. The preliminary analysis on uncertainty as a quality signal (Section 3.3) is insightful, justifying the method's design choices. The paper clearly articulates the problem, the proposed solution, and the significance of the results.
- Potential for broader impact and future work: The Adv-RM framework provides a new tool for probing and understanding RM weaknesses. The findings and the proposed method open avenues for further research into more robust alignment techniques. The authors' commitment to open-sourcing code and data will facilitate reproducibility and further community engagement.

**Reasons To Reject:**

- Specificity of robustness gains. The paper demonstrates that Adv-RMs become more robust to the specific type of adversarial attack used to generate $y_{adv}$. While Figure 6a shows robustness against repeated attacks of the same kind, the extent to which this robustness generalizes to other unseen types of OOD inputs or different reward hacking strategies is not extensively explored. The limited attack transferability shown in Figure 3 might suggest the learned robustness is somewhat narrow. A critic might argue for more diverse robustness evaluations.
- Dependence on the quality/nature of $R_{θ2}$. The effectiveness of the adversarial example generation depends on having a suitable $R_{θ2}$ that provides meaningful disagreement. If $R_{θ2}$ is poorly chosen (e.g., too similar, too dissimilar, or low quality itself), the generated adversarial examples might not be effective or representative of true vulnerabilities. The paper uses strong RMs for $R_{θ2}$ but doesn't deeply analyze the sensitivity to this choice, which could be a practical limitation for users without access to multiple high-quality RMs.
- Computational cost as a significant barrier. A 3x increase in computational cost for RLHF is substantial. While the authors argue the robustness gains justify this, for many researchers or smaller labs, this could be a prohibitive barrier to adoption. The paper acknowledges this but doesn't offer clear pathways to significantly mitigate this cost while retaining most benefits, which might limit its practical impact for a broader audience.
- Modest gains on static benchmarks / peak performance. While stability is improved, the gains in final SFT policy performance on static benchmarks (e.g., RewardBench scores in Table 4, peak Gold Reward in Figure 4c) are relatively modest and have more variance than RRM. A reviewer focused solely on pushing SOTA performance metrics might see this as a limitation, even if stability is a valuable contribution in itself. The argument could be made that the method primarily prevents degradation rather than significantly elevating best-case performance.
- Limited exploration of "low quality" characterization. The paper states $y_{adv}$ are "OOD and of low quality." OOD is measured, but "low quality" is more of an emergent property from $R_{θ1}$ liking it and $R_{θ2}$ disliking it. A deeper analysis or characterization of the types of low-quality behaviors discovered could strengthen the paper.

It's important to note that many of these points are about the degree of contribution or exploration rather than fundamental flaws.

---

> ### Author Response · Authors · 2025-05-30
> **Response to Reviewer qJrb**
>
> Dear Reviewer qJrb,
>
> Thank you for your detailed and insightful review! We are glad you think our paper makes several strong contributions. We address your review below.
>
> **Reason to Reject 1: Specificity of robustness gains.**
> - We agree this is an important aspect to study. However, as this is one of the first works to attack RMs, there are limited attack strategies we can try. In the paper we try Textfooler (which does not work well on RMs) as well as transfer attacks from hacking other RMs. Recently, reWordBench showed how we can perturb some existing benchmarks to attack RMs, but the authors do not publicly release their data or code. However, if any other new attack method comes out, we will try it with our method. In general, we believe AdvRM gives robustness to new types of OOD data, largely because it helps mitigate reward hacking.
>
> **Reason to Reject 2: Dependence on $R_{\theta_2}$.**
> - Indeed, the dependence on $R_{\theta_2}$ somewhat restricts the nature of our adversarial attacks, and we believe that finding a better choice of $R_{\theta_2}$ is an important topic to study in future work. However, the reliance on $R_{\theta_2}$ is not necessarily a practical issue, as we show simply choosing $R_{\theta_2}$ to be the same architecture as the targeted RM but with a different training seed is sufficient for good performance. Users can always access such a checkpoint and therefore can use AdvRM.
>
> **Reason to Reject 3: Computational cost.**
> - We agree this is a drawback of our method. We do not investigate reducing the training cost, but believe there should be several straightforward ways to do so. In our paper we use llama 3.1 8B as the adversarial policy, and find that it is capable of hacking much larger RMs (i.e. 340B parameters). This makes us optimistic that even smaller models could be used as the adversarial policy, such as Qwen-1.5B, or that parameter-efficient fine-tuning techniques could be used. These approaches would significantly reduce the training cost.
>
> **Reason to Reject 4: Gains in peak performance.**
> - This is a good point. The main focus of this paper is not on improving peak performance, but rather on (1) investigating whether adversarial examples exist for RMs and (2) using such adversarial examples to improve RM robustness. Although AdvRM only slightly improves peak performance, it gives us insight into RM vulnerabilities, stabilizes training, and can hopefully prevent unexpected behaviors from arising during training. Robustness has always been valued by the ML community, and we believe that robustness is important for LLM systems.
>
> **Reason to Reject 5: Limited exploration of low quality characterization.**
> - The phenomenon that OOD samples generated from generative models are low quality has been found by many papers [1, 2, 3], and it is from these papers where we get a lot of our intuition. We will cite these papers and explain their findings in our paper.
> 1. Holtzman et al., "The Curious Case of Neural Text Degeneration", ICLR 2020
> 2. Ouyang et al., "Training Language Models to Follow Instructions with Human Feedback", NeurIPS 2022
> 3. Kingma & Welling, "Auto-Encoding Variational Bayes", arXiv 2013
> - We conduct further study on this in section 3.3, where we indeed find that OOD samples are low quality. This matches our intuition that OOD samples are low quality because they deviate from the expert distribution. We leave more in-depth study of this phenomenon to future work.

---

> > ### Author Response · Authors · 2025-05-30
> > **Rebuttal Continued**
> >
> > **Question 1: In-house RM details.**
> > - This in-house RM is a Llama 3.1 70B RM trained with a regression style loss on the helpsteer2-preference data. So compared to the Llama-Nemotron-3.1-Reward, it has the same architecture and training data but it is trained with a slightly different objective.
> > It is hard to quantitatively measure the importance of R_theta_2 but in general we find that the choice of R_theta_2 does matter but it is not the sole decider of the adversarial text. For example, when we attack the 340B RM with the Llama-Nemotron-3.1-Reward, the responses become stylistically very bland. We hypothesize that this is because R_theta_2 (Llama-Nemotron-3.1-Reward) cares a lot about style and therefore the adversarial policy generates responses with poor style. When we attack the Skyworks RM with Llama-Nemotron-3.1-Reward however, the responses are a little stylistically bland, but not to the same degree as with the 340B RM. Therefore the choices of both R_theta_1 and R_theta_2 impact the downstream attacks.
> >
> > **Question 2: unseen OOD inputs.**
> > - Please see our response to reason to reject 1.
> >
> > **Question 3: Adversarial Robustness.**
> > - The reason why we fail to find adversarial examples is because the policy model cannot find samples that are OOD yet receive high reward. We will add the training plot to the next version of the paper, but in essence it shows a flat reward line, as the agent learns nothing. This implies both of your statements are true: the reward model becomes hardened against any OOD samples the policy can find, but this does not imply it is robust against all attacks. If we use more advanced exploration strategies or use more compute in the exploration process, we can probably find more adversarial examples. These results imply the iterative adversarial training process is limited by the exploration capabilities of the attack policy.
> >
> > **Question 4: Computational cost.**
> > - We are quite optimistic that a lightweight adversarial training could perform well (maybe using 1.5B models or LoRA), therefore reducing training cost. This is because we can use an 8B policy to successfully attack a 340B policy. However, we hypothesize that a larger attack policy would be more capable of finding adversarial examples than a small one. We leave such study to future work.
> >
> > **Question 5: Characteristics of adversarial examples.**
> > - We find that the attack text varies drastically for different RMs (see Figure 2), and that the attack characteristics are dependent on both the $R_{\theta_1}$ and $R_{\theta_2}$ (see line 263 for more details). Interestingly, we find that the attack policy usually converges to a fixed attack strategy, such as responses with no punctuation for the 340B RM. We will add this detail to the next version of the paper.
> >
> > **Question 6: diagnosing blind spots in RMs.**
> > Yes this is a great point! When reward models are being used to moderate LLM responses [1], finding their vulnerabilities becomes a task of utmost importance. AdvRM can be used to red-team these systems. We will emphasize this in the text.
> > 1. Inan et al., "LLaMA Guard: LLM-Based Input-Output Safeguard for Human-AI Conversations", arXiv (2023)
> >
> > Thank you again for your in-depth review of our paper. Please let us know if you have any further questions, and we look forward to more discussion!

---

> > > ### Comment · Reviewer_qJrb · 2025-06-04
> > >
> > > Thank you for your response. I maintain my score.

---

> > > > ### Author Response · Authors · 2025-06-06
> > > > **Thank you for the discussion!**
> > > >
> > > > Thank you again for the detailed review! Please let us know if you have any further questions.

---

### Official Review · Reviewer_3VZA · 2025-05-12

**Rating:** 6
**Confidence:** 4
**Ethics Flag:** 1

**Summary:**

This paper investigates the robustness of reward models (RMs), majorly exploring two questions:

(i) how to measure the robustness of RMs and (ii) how to improve the robustness.

The authors formulates the robustness problem by attacking the reward model, where an adversarial policy is trained to optimize the reward gap between the target RM (R_1) and the other RM model's rating(R_2): $$ R_1 (x, y) - \lambda R_2  (x, y)$$ satisfying $R_1(x,y) > \tau$ and $\tau$ is a threshold.

These pairs are filter to form an adversarial training set for enhancing the RM, and the RM is improved iteratively as the RLHF training goes.

Evaluation with synthetic dataset and real-world RLHF examples show that the proposed method could identify pairs that can successfully attack current models and the adversarial training improves the robustness.


## After Rebuttal
The authors response effectively addressed my concerns regarding the motivation and experiments.

**Questions To Authors:**

- Line 197 states that high quality responses are adopted in the preference training against the OOD pair, would the replay of high-quality responses yield performance gain? An ablation for this might be interesting, i.e., performing SFT with those chosen pairs.

- Line 518: More details are needed for the human evaluation part. How many samples went through  the human evaluation and what's the consistency between human annotators?
- Table 4 shows that the improvements on  rewardbench is rather marginal and  even worse on Chat and Safety, this is confusing and it indicates that the adversarial training might be useless for in real-world scenarios? Any explanations for the degraded performance and what's the performance of RRM?

- As mentioned in Line 295, RRM and the proposel are complementary, would the combination leads to better results?

**Reasons To Accept:**

- Overall, this paper is well-written with clearly defined goal and well well-executed experimental setup.

- The reward hacking problem is of great significance for LLM's alignment.

- The proposed method using the gap between two reward models for identifying OOD responses is somewhat novel and and the illustrated examples seem to be interesting and transferable.

**Reasons To Reject:**

- The utility of adversarial training is not clear: reward hacking is a big issue, while the investigated adversarial robustness might not. Since the responses identified are OOD, they would rarely appear during the RLHF training. Therefore, these adversarial examples are not the root of reward hacking.
- The experimental results are mainly conducted on a small dataset consisting of 128 prompts, which is not convincing despite the clear gap. I am wondering whether the distribution would be narrow, and suggest the authors could evaluate on a larger evaluation set (> 500 prompts) for more comprehensive evaluation. The results on RewardBench highlight this concern as well.
- Seems the baseline RRM is sufficiently good which simply samples responses from other prompts in both synthetic and real-world datasets and this renders advantages of the proposal to be unclear given the computation overhead.

---

> ### Author Response · Authors · 2025-05-30
> **Response to Reviewer 3VZA**
>
> Dear Reviewer 3VZA,
>
> Thank you for your detailed and insightful review! We are glad you think our paper is well written and novel. The reviewer's main concern seems to be that RRM already provides a strong baseline, and therefore our contribution may be limited. However, we have many contributions when compared to RRM, including showing the existence of adversarial examples for SOTA reward models (which has not been shown before), providing an approach to find such examples, and studying the properties of such adversarial examples. Moreover, AdvRM outperforms RRM on several downstream tasks while using a completely new approach, which can be built upon by future works. Therefore, we feel our paper still has several meaningful contributions to the COLM community.
>
> For the remaining concerns, we believe we can address them in the next version of the paper. We leave our rebuttal to your review below.
>
> **Reason to Reject 1: The utility of adversarial examples is unclear.**
> - The connection between OOD samples and reward hacking in RLHF is a well studied phenomenon. Concretely, [1] and [2] both show that reward hacking occurs due to poor performance on OOD data. Therefore, the adversarial samples we find directly address the root of the reward hacking problem. We will discuss these findings in the next version of our paper.
> 1. Eisenstein et al., "Helping or herding?" COLM 2024
> 2. Coste et al., "Reward Model Ensembles Help Mitigate Overoptimization", ICLR 2024
> - Moreover, in this paper we show there is a clear empirical benefit of adversarial training compared to conventional RLHF training, with the adversarial trained models achieving better downstream performance and more stable training.
>
> **Reason to Reject 2: The experimental results are mainly conducted on a small dataset of 128 prompts.**
> - We want to clarify that we have two different experimental setups: one where we train on 10,000 synthetic preference pairs and one where we train on 10,000 real preference pairs given by human annotators. In both settings, we evaluate performance across a test set of 128 prompts as well as a set of 128 prompts from the training dataset. This is in line with prior works on LLM alignment: MT-Bench uses 80 conversations, the Koala benchmark has 180, Vicuna’s eval has 80 samples, and WizardLM eval set has 218 samples.
> - In addition, we conduct our experiments with three random seeds in the synthetic setting. We can use these runs to conduct a two sided t-test, in which we can confirm the difference between AdvRM and conventional RLHF is statistically significant.
> Finally, we supplement our evaluation to make the set of prompts have size 512. The results (in the synthetic setting, corresponding to Figure 4c) can be found below. Our results show that performance is consistent with larger test set sizes.
>
> | Method           | Test Performance (Mean ± Std) |
> |------------------|-------------------------------|
> | AdvRM            | 7.15 ± 0.66                   |
> | RRM              | 7.01 ± 0.34                   |
> | Ens (std)        | 6.17 ± 0.14                   |
> | Ens (mean)       | 5.96 ± 0.60                   |
> | RLHF (baseline)  | 2.86 ± 2.52                   |

---

> > ### Author Response · Authors · 2025-05-30
> > **Rebuttal Continued**
> >
> > **Reason to Reject 3: The RRM baseline is sufficiently good.**
> > - Indeed, RRM is a strong baseline. However, we remark that AdvRM outperforms RRM, allowing for stable training for 30% longer than RRM (see Figure 5).
> > - In addition, AdvRM explicitly finds adversarial samples for the targeted reward model. This provides interpretability regarding the vulnerabilities of the reward model, which is especially useful when reward models are being used as moderation tools. RRM on the other hand does not find adversarial examples (see Table 1 of the paper).
> > - Finally, we remark that the computational cost of RRM is not necessarily lower than that of AdvRM. In the original paper, RRM increases the dataset size anywhere from 3 to 14 times, which increases the cost of reward model training from 3 to 14 times. When the reward model size is large the reward model training cost is large (i.e. 256 H100 hours for a 70B model), so such large increases in dataset size are very costly. On the other hand, with AdvRM we can attack large RMs with a small policy model (we attack a 340B RM with a 8B policy) and AdvRM only increases the RM dataset size by 10%, meaning the cost of AdvRM remains approximately constant even if we increase RM size. Therefore with very large RMs, AdvRM can be much cheaper than RRM.
> >
> > **Question 1: Would the replay of high quality samples result in performance gain?**
> > - Thank you for the suggestion! We have conducted the ablation. We conduct additional SFT on this data, whose definition can be seen on line 198. It is data sampled from the SFT policy that has higher than average reward, so directly SFTing on it is equivalent to a form of rejection sampling. The results can be seen in the below table, where we use the same setting as Figure 5a (llama 405B as judge). We tune the SFT learning rate in [1e-6, 2e-6, 5e-6], select the one with the highest winning rate, and denote it as “SFT-rejection sampling.” We find that this baseline does slightly improve performance over the baseline, but not nearly as much as conventional RLHF. Moreover, it is unclear how these higher quality samples affect robustness to reward hacking, which is the main goal of our method.
> >
> > | Training Approach         | Win Rate (%) |
> > |---------------------------|--------------|
> > | SFT-rejection sampling    | 67.2         |
> > | Conventional RLHF         | 78.1         |
> > | Ens-Mean                  | 80.1         |
> > | Ens-Std                   | 76.5         |
> > | RRM                       | 79.8         |
> > | AdvRM                     | 80.9         |
> >
> > **Question 2: More details are needed for human evaluation.**
> > - Thank you for the suggestion on this. In total the human annotators evaluate 479 pairs. Note that this is the same amount as other alignment works, such as DPO. We evaluate annotator agreement over 133 samples, and find a 79.12 agreement rate. The reason for this high agreement rate is that we tell annotators to mark pairs as a tie if the two samples are of fairly similar quality, and only say one response wins if it is significantly better than another response. This eliminates a lot of ambiguity from the annotation process. We will include these details in the next version of the paper.
> >
> > **Question 3: Why only marginal improvements on rewardbench?**
> > - It is a well-known tradeoff in machine learning that adversarial training entails a tradeoff: adversarial training increases robustness, but often at the expense of average-case performance. This could account for some small degradation in performance. Despite this tradeoff, we find that AdvRM can improve both robustness (to AdvRM attacks and reward hacking) while maintaining or slightly improving average case performance (reward bench accuracy).
> > - We believe the statement that our method is “useless in real-world scenarios” overlooks several important contributions of our work. In addition to improving RewardBench performance, our method (1) provides interpretability into reward model weaknesses (Figure 2), (2) enables more stable and performant RLHF training—the primary use case for reward models (Figures 4 and 5), and (3) significantly improves robustness to adversarial attacks (Figure 6).
> > - We show the RRM scores on RewardBench below. As we can see, RRM decreases RewardBench scores, and is significantly worse than AdvRM. We hypothesize this is due to training too much on the relatively easy RRM data. We will include this baseline in the next revision of the paper.
> >
> >
> > | Method   | Overall | Chat   | Chat Hard | Safety |
> > |----------|---------|--------|-----------|--------|
> > | AdvRM    | 83.99   | 96.09  | 64.04     | 85.14  |
> > | Baseline | 83.29   | 97.21  | 61.18     | 85.41  |
> > | RRM      | 82.55   | 94.13  | 68.42     | 82.57  |

---

> > > ### Author Response · Authors · 2025-05-30
> > > **Rebuttal Continued**
> > >
> > > **Question 4: RRM and AdvRM are complementary, would the combination lead to better results?**
> > > - Yes, we believe they should be complementary. However, combining RRM and AdvRM requires significant computational resources to find the right data blend and run our large scale RLHF jobs. Therefore, we leave such study to future work.
> > >
> > >
> > > Thank you again for your detailed review. By including the new studies and discussion into our paper, it is substantially strengthened. We look forward to further discussion!

---

> > > > ### Comment · Reviewer_3VZA · 2025-06-06
> > > >
> > > > Thank you for your response. The clarifications and new results (statistical test, comparison v.s. SFT RFT) effectively address my concerns. I will increase my scores accordingly.
> > > >
> > > > Also, there is a new version of RewardBench (RewardBench 2: Advancing Reward Model Evaluation), and incorporating the results might introduce more insights.

---

> > > > > ### Author Response · Authors · 2025-06-06
> > > > > **Thank you for the discussion!**
> > > > >
> > > > > Thank you again for the rigorous review and for being willing to increase your score! We will be sure to include the discussed changes into our paper.
> > > > >
> > > > > Thanks for letting us know about RewardBench 2 as well. We will investigate how our trained models do on it.

---

### Decision · Program_Chairs · 2025-07-08

**Decision:**

Accept

**Comment:**

The authors study the problem of robustness of reward models used in language model alignment, particularly robustness with respect to out of distribution/low quality inputs. The authors develop a novel adversarial training technique and demonstrate that the approach is effective across several benchmarks. Some reviewers raised concerns, particularly on the comprehensiveness of the evaluations used in the paper and demonstrating that the findings scale to larger reward and language models. These were addressed to an extent in the rebuttal, and hence I recommend acceptance, while encouraging the authors to address the concerns in the camera ready version of the paper.

[Automatically added comment] At least one review was discounted during the decision process due to quality]